# Robust Neural ODEs via Contractivity-promoting regularization

## Abstract

Neural networks can be fragile to input noise and adversarial attacks. In this work, we consider Neural Ordinary Differential Equations (NODEs) – a family of continuous-depth neural networks represented by dynamical systems - and propose to use contraction theory to improve their robustness. A dynamical system is contractive if two trajectories starting from different initial conditions converge to each other exponentially fast. Contractive NODEs can enjoy increased robustness as slight perturbations of the features do not cause a significant change in the output. Contractivity can be induced during training by using a regularization term involving the Jacobian of the system dynamics. To reduce the computational burden, we show that it can also be promoted using carefully selected weight regularization terms for a class of NODEs with slope-restricted activation functions, including convolutional networks commonly used in image classification. The performance of the proposed regularizers is illustrated through benchmark image classification tasks on MNIST and FashionMNIST datasets, where images are corrupted by different kinds of noise and attacks.

## 1 Introduction

Neural networks (NNs) have demonstrated outstanding performance in image classification, natural language processing, and speech recognition tasks. However, they can be sensitive to input noise or meticulously crafted adversarial attacks (Xu et al., 2020; Carlini & Wagner, 2017; Athalye et al., 2018; Szegedy et al., 2013). The customary remedies are either heuristic, such as feature obfuscation (Miller et al., 2020), adversarial training (Goodfellow et al., 2014; Allen-Zhu & Li, 2022), and defensive distillation (Papernot et al., 2016), or certificate-based such as Lipschitz regularization (Xu et al., 2020; Fazlyab et al., 2019; Pauli et al., 2021; Aquino et al., 2022; Virmaux & Scaman, 2018; Combettes & Pesquet, 2020). The overall intent of certificate-based approaches is to penalize the input-to-output sensitivity of NNs to improve robustness.

Recently, the connections between NNs and dynamical systems have been extensively explored. Representative results include classes of NNs stemming from the discretization of dynamical systems (Haber & Ruthotto, 2017) and NODEs (Chen et al., 2018), which transform the input through a continuous-time ODE embedding training parameters. The continuous-time nature of NODEs makes them particularly suitable for learning complex dynamical systems (Rubanova et al., 2019; Greydanus et al., 2019) and allows borrowing tools from dynamical system theory to analyze their properties (Fazlyab et al., 2022; Galimberti et al., 2021).

In this paper, we employ contraction theory to improve the robustness of NODEs. A dynamical system is contractive if all trajectories converge exponentially fast to each other (Lohmiller & Slotine, 1998; Tsukamoto et al., 2021). Through the lens of contraction, slight perturbations of initial conditions have a diminishing impact over time on the NODE state. With the above considerations, we propose a class of regularizers that promote contractivity of NODEs during the training. In the most general case, the regularizers require the Jacobian matrix of the NODE, which might be computationally challenging to obtain for deep networks. Nevertheless, for a wide class of NODEs with slope-restricted activation functions, we show that contractivity can be promoted by directly penalizing the weights during the training. Moreover, by leveraging the linearity of convolution operations, we demonstrate that contractivity can be promoted for convolutional NODEs by regularizing the convolution filters only.

## 1.1 RELATED WORK

Several works have focused on improving the robustness of general NNs against input noise and adversarial attacks using dynamical system theory. For example, the notion of incremental dissipativity is used to provide robustness certificates for NNs in the form of a linear matrix inequality (Aquino et al., 2022). The works Chen et al. (2021; 2022) address the robustness issue of NNs by using a closed-loop control method from the perspective of dynamical systems. A control process is added to a trained NN to generate control signals to mitigate the perturbations in input data. Nevertheless, the method requires to solve an optimal control problem for the inference of an input sample, which increases the computational burden.

A detailed study on the robustness of NODEs has been done by Hanshu et al. (2019), where the authors show that NODEs can be more robust against random perturbations than common convolutional NNs. Moreover, they study time-invariant NODEs, and propose to regularize their flows to further enhance the robustness. To bolster the defense against adversarial attacks, NODEs equipped with Lyapunov-stable equilibrium points have been proposed (Kang et al., 2021). Likewise, Rodriguez et al. (2022) introduced a loss function to promote robustness based on a control-theoretic Lyapunov condition. Both methods have shown promising performance against adversarial attacks. Finally, Massaroli et al. (2020) design provably stable NODEs and argue that stability can reduce the sensitivity to small perturbations of the input data. Nevertheless, this claim is not supported by theoretical analysis or numerical validation. In comparison to all the aforementioned works, in this paper, we employ contraction theory to regularize the trajectories of NODEs and improve robustness.

Recently, contraction theory has been employed in the framework of NNs for various purposes. For instance, contractivity is exploited to improve the well-posedness and robustness of implicit NNs (Jafarpour et al., 2021), the trainability of recurrent NNs (Revay & Manchester, 2020; Jafarpour et al., 2022), and the analysis of Hopfield NNs with Hebbian learning (Centorrino et al., 2022). In Zakwan et al. (2022), the authors propose a Hamiltonian NODE that is contractive by design to improve robustness. However, the extension to different classes of NODEs, including convolutional NODEs, is not straightforward. Besides the robustification of NNs and NODEs, contractivity has also been exploited for learning NN-based dynamical models from data. For instance, Singh et al. (2021) and Revay et al. (2021a;b) utilize contraction theory to learn stabilizable nonlinear NN models from available data.

## 1.2 CONTRIBUTIONS

The contribution of this paper is fourfold.

- We show that contractivity can be used to improve the robustness of NODEs, and demonstrate how to promote contractivity for general NODEs during training by including regularization terms in the cost function.

- The regularization terms involve optimizing the Jacobian matrix in NODEs, which might be computationally expensive. Interestingly, for a wide class of NODEs with slope-restricted activation functions, we prove that contractivity can be promoted by carefully penalizing weight matrices and without optimizing the Jacobian matrix.

- By exploiting the linearity of convolution operations and the above results for NODEs with slope-restricted activation functions, we show that contractivity for convolutional NODEs can be induced by suitably penalizing the convolutional filters.

- We conduct experiments on MNIST and FashionMNIST datasets with test images perturbed by different kinds of noise and adversarial attacks. Compared to vanilla NODEs, by using contractivity-promoting regularization terms the average test accuracy can be improved up to 34% in the presence of input noise and up to 30% in the case of adversarial attacks.

### 1.3 ORGANIZATION AND NOTATION

The paper is organized as follows: Section 2 provides preliminaries on NODEs and contraction theory. In Section 3, we propose several regularization approaches for NODEs to promote contractivity. Numerical experiments are described in Section 4, and Section 5 concludes the paper.

The set of real numbers is $\mathbb{R}$. $\frac{\partial f(x)}{\partial x}$ represents the Jacobian matrix of a continuously differentiable function $f(\cdot)$. The minimal eigenvalue of a symmetric matrix $A$ is denoted as $\lambda_{\min}(A)$. $\mathrm{diag}(x)$ represents a diagonal matrix with the entries of the vector $x$ on the diagonal. For symmetric matrices $A$ and $B$, $A \succ (\succeq)B$ means that $A - B$ is positive (semi)definite. $I$ denotes the identity matrix. The 2-norm is denoted as $\|\cdot\|$.

## 2 PRELIMINARIES

### 2.1 NEURAL ORDINARY DIFFERENTIAL EQUATION

A NODE is represented by the dynamical system

$$\dot{x}_t = f(x_t, \theta_t, t), \quad t \in [0, T] , \tag{1}$$

where $x_t \in \mathbb{R}^n$ is the state of the NODE and $f(x_t, \theta_t, t)$ is a generic smooth function with parameters $\theta_t \in \mathbb{R}^m$. When used in machine learning tasks, the NODE is usually pre- and post- pended with additional layers, e.g., $x_0 = h_\alpha(z)$ and $y = g_\beta(x_T)$, where $h_\alpha, g_\beta$ are NNs with parameters $\alpha \in \mathbb{R}^{n_\alpha}, \beta \in \mathbb{R}^{n_\beta}$, respectively, $z \in \mathbb{R}^{n_z}$ is the input feature, $y \in \mathbb{R}^p$ represents the output, and $x_0, x_T$ are the state of the NODE (1) at time $t = 0$, and $t = T$, respectively.

Several methods have been proposed for training NODEs, such as the adjoint sensitivity method (Chen et al., 2018), and the auto-differentiation technique (Paszke et al., 2017). In this paper, we use the most straightforward approach, that is, the time-discretization of (1) (Haber & Ruthotto, 2017). Consider a classification task, and suppose the training dataset is $\{z_i, c_i\}_{i=1}^s$, where $z_i$ are the input features (e.g. images), $c_i$ are the corresponding labels, and $s$ is the number of training samples. Before training, the NODE (1) is discretized and the resulting discrete-time equations define each of the network layers. For instance, by using Forward Euler (FE) method one obtains[1]

$$x_{k+1} = x_k + hf(x_k, \theta_k, k), \quad k = 0, \dots, \frac{T}{h} - 1 , \tag{2}$$

where $h > 0$ is the sampling period. Then, the NODE is trained by solving the optimization problem

$$\min_{\alpha, \{\theta_k\}_{k=0}^{T/h-1}, \beta} \quad \sum_{i=1}^s l(y_i, c_i) + \gamma \mathrm{reg}(\alpha, \{\theta_k\}_{k=0}^{T/h-1}, \beta) \tag{3}$$

$$\text{s.t.} \quad x_0^i = h_\alpha(z_i), \quad i = 1, \dots, s ,$$

$$x_{k+1}^i = x_k^i + hf(x_k^i, \theta_k, k), \quad k = 0, \dots, \frac{T}{h} - 1, \tag{4}$$

$$y_i = g_\beta(x_{T/h}^i) ,$$

where $l(\cdot, \cdot)$ denotes the loss function, and $\mathrm{reg}(\cdot)$ is a regularization term suitably scaled by the regularization parameter $\gamma > 0$. For brevity, throughout the paper, we omit the pre- and post-pended layers $h_\alpha(\cdot)$ and $g_\beta(\cdot)$, which usually depend on the specific learning task (Chen et al., 2018).

### 2.2 CONTRACTIVITY

Contractivity is a property of dynamical systems, and it implies that the trajectories of the dynamical system converge to each other asymptotically. The formal definition is given below.

**Definition 1.** *The dynamics (1) is contractive with a contraction rate $\rho > 0$ if*

$$\|\hat{x}_t - x_t\| \le e^{-\rho t}\|\hat{x}_0 - x_0\|, \quad \forall t \in [0, T] , \tag{5}$$

*for all $x_0, \hat{x}_0 \in \mathbb{R}^n$, where $\hat{x}_t$ and $x_t$ are the solutions of (1) with initial conditions $\hat{x}_0$ and $x_0$, respectively.*

---

[1]For simplicity, we assume that $\frac{T}{h}$ is an integer.

Therefore, if a NODE is contractive, the Lipschitz constant between the input and the output is smaller than 1, that is, $\frac{\|\hat{x}_T - x_T\|}{\|\hat{x}_0 - x_0\|} < 1$ for any $\hat{x}_0, x_0$. As a result, contractive NODEs are robust in the sense that a slight perturbation in the input features $x_0$ would not result in a large deviation in the output $x_T$. Moreover, we have that the NODE (1) is contractive with a contraction rate $\rho$, if and only if (Tsukamoto et al., 2021)

$$-\rho I - \left( \frac{\partial f}{\partial x} + \frac{\partial f}{\partial x}^\top \right) \succ 0, \quad \forall t \in [0, T], x \in \mathbb{R}^n \,, \tag{6}$$

where $\frac{\partial f}{\partial x}$ is the Jacobian matrix of $f$.

**Remark 1.** *The notion of asymptotic stability used in Massaroli et al. (2020) might not be appropriate for promoting robustness of NNs. Indeed, as shown in Rüffer et al. (2013), although for convergent dynamics the perturbed states eventually converge to a unique trajectory, after a finite time, the distance between trajectories can be arbitrarily large, which can result in poor robustness of the NODE (1). In contrast, contractive dynamics does not suffer from this problem, and we will show in Section 4 that contractivity can considerably improve the robustness of NODEs.*

**Remark 2.** *Contractivity implies all the state trajectories of (1) converge exponentially fast to an equilibrium (Tsukamoto et al., 2021), which may limit the representation power of NODEs. However, a loss of expressivity might be unavoidable for increasing robustness, as discussed in Tsipras et al. (2019).*

**Remark 3.** *When training NODEs with global contractivity requirement, the training time $T$ is finite, and we can also tune the contraction rate $\rho$, which is a hyper-parameter. As a result, the NODE trajectory would neither diverge nor converge to the same point during training, which ensures good learning and robustness performance. The readers can also refer to Figure 1 in Zakwan et al. (2022) as an illustration showing that global contraction can still ensure good learning result.*

## 3 CONTRACTIVITY-PROMOTING REGULARIZATION

To promote the robustness of the NODE (1), one can leverage a regularization term penalizing the violation of (6). Contractivity requires the inequality (6) to hold for all $t \in [0, T], x \in \mathbb{R}^n$. However, during the training, we only have access to discretized states $x_k^i$ and hence, we can promote the fulfillment of the condition (6) by using the following regularization term in (3)

$$\text{reg}(\{\theta_k\}_{k=0}^{T/h-1}) = \sum_{i=1}^{s} \sum_{k=0}^{T/h} \text{ReLU}\left( -\lambda_{\min}\left( -\rho I - \left( \frac{\partial f}{\partial x} + \frac{\partial f}{\partial x}^\top \right)\Big|_{x_k^i, k} \right) \right), \tag{7}$$

where $\text{ReLU}(\cdot)$ denotes the ReLU activation function.

**Remark 4.** *Although the regularizer (7) stems from (6), there are some differences. The condition (6) implies that all the trajectories converge to each other exponentially fast. In contrast, the regularizer (7) only penalizes the violation of contractivity locally on the sampled state $x_k^i$, which is weaker than (6) and therefore imposes fewer constraints on NODEs. Due to the smoothness property of NODEs, one can show that the learned trajectories $\{x_0^i, x_1^i, \ldots, x_{T/h}^i\}$ are locally contractive in the sense that the relation (5) holds only in the neighborhood of $x_k, k = 0, \ldots, T/h$. We defer the reader to Section 4.1 for an illustration showing the benefits of using (7).*

### 3.1 WEIGHT REGULARIZATION FOR IMPROVING TRAINING COMPLEXITY

Since the regularization term (7) involves the Jacobian matrices $\frac{\partial f}{\partial x}|_{x_k^i, k}$ for all $i, k$, it might be computationally expensive to obtain. In this section, we focus on a family of NODEs with slope-restricted activation functions and show that one can directly regularize their trainable parameters to promote contractivity. Consider the following NODE

$$\dot{x}_t = \sigma(W_t x_t + b_t), \quad t \in [0, T] \,, \tag{8}$$

where $x_t \in \mathbb{R}^n$ is the state, $W_t \in \mathbb{R}^{n \times n}, b_t \in \mathbb{R}^n$ are NN parameters, and $\sigma(\cdot)$ is the activation function. The following theorem provides a sufficient condition on the weights $W_t$ guaranteeing that (8) is contractive.

**Theorem 1.** *Assume $\sigma'(\cdot) \in [\underline{\kappa}, \bar{\kappa}]$, where $\sigma'(\cdot)$ denotes any sub-derivative of $\sigma$, and $\bar{\kappa} > \underline{\kappa} > 0$. Moreover, for $\rho > 0$, let the following condition hold*

$$-\rho - 2\underline{\kappa}W_{t,ii} - \bar{\kappa} \sum_{j=1, j\neq i}^{n} (|W_{t,ij}| + |W_{t,ji}|) > 0, \quad i = 1, \ldots, n \tag{9}$$

*for $t \in [0, T]$, where $W_{t,ij}$ is the $ij$-th element of $W_t$. Then, the NODE (8) is contractive with a contraction rate $\rho$.*

*Proof.* From (6), the NODE (8) is contractive with a contraction rate $\rho$ if

$$-\rho I - J_t W_t - W_t^\top J_t \succ 0, \quad \forall x \in \mathbb{R}^n, t \in [0, T] , \tag{10}$$

where $J_t$ is the Jacobian matrix of $\sigma(W_t x_t + b_t)$ with respect to the input $W_t x_t + b_t$. It follows that $J_t$ is a diagonal matrix with the $i$-th diagonal entry equal to $\sigma'([W_t x_t + b_t]_i)$, where $[W_t x_t + b_t]_i$ denotes the $i$-th element of $W_t x_t + b_t$. According to the Gersgorin disk theorem (Horn & Johnson, 1985), any matrix $S \in \mathbb{R}^{n \times n}$ that satisfies the following conditions

$$S_{ii} > \sum_{j=1, j\neq i}^{n} |S_{ij}|, \quad i = 1, \ldots, n$$

is positive definite (i.e. $S \succ 0$). The diagonal elements of the matrix $-\rho I - JW - W^\top J$ (where the subscript $t$ is dropped for simplicity) are

$$-\rho - 2J_{ii}W_{ii} ,$$

where $J_{ii}, W_{ii}$ are the $ii$-th elements of the matrices $J$ and $W$, respectively. Moreover, the $ij$-th $(i \neq j)$ elements of the matrix $-\rho I - JW - W^\top J$ are

$$-J_{ii}W_{ij} - J_{jj}W_{ji} .$$

Therefore, in view of Gersgorin disk theorem, the matrix $-\rho I - JW - W^\top J$ is positive definite if

$$-\rho - 2J_{ii}W_{ii} > \sum_{j=1, j\neq i}^{n} |J_{ii}W_{ij} + J_{jj}W_{ji}|, \quad i = 1, \ldots, n . \tag{11}$$

A sufficient condition for the feasibility of (11) is that the lower bound of the LHS is greater than the upper bound of the RHS. Consequently, it is necessary that $W_{ii} \leq 0$. Since $\bar{\kappa} \geq \sigma'(\cdot) \geq \underline{\kappa}$, a lower bound of the LHS of (11) is

$$-\rho - 2\underline{\kappa}W_{ii} ,$$

and an upper bound of the RHS of (11) is

$$\bar{\kappa} \left( \sum_{j=1, j\neq i}^{n} |W_{ij}| + |W_{ji}| \right) .$$

Hence, if the condition (9) holds for all $i$ and $t \in [0, T]$, the inequality (10) is verified, and the NODE (8) is contractive with the contraction rate $\rho$. □

Inspired by the above result, we can use the following regularization term in (3) during the training to promote contractivity of the NODE (8)

$$\text{reg}\left(\{W_k\}_{k=0}^{T/h-1}\right) = \sum_{k=0}^{T/h-1} \sum_{i=1}^{n} \text{ReLU}\left(\rho + 2(\underline{\kappa} + \bar{\kappa})W_{k,ii} + \bar{\kappa} \sum_{j=1}^{n} (|W_{k,ij}| + |W_{k,ji}|)\right) , \tag{12}$$

where $W_k$ is the discretized counterpart of $W_t$ during the training.

**Remark 5.** *Similar to the Hamiltonian NODEs in Zakwan et al. (2022) ensuring contractivity by design, in view of Theorem 1, one can parameterize a subset of the weight matrices of NODE (8) that satisfy the condition (9) by design. The main idea is to modify the diagonal elements of $W_t$ such that the resulting weight matrices $\tilde{W}_t$ satisfy (9) automatically. These matrices can be written as*

$$\tilde{W}_t = W_t + H_t ,$$

*where $H_t = \text{diag}(H_{t,1}, \ldots, H_{t,n})$ with $2\underline{\kappa}H_{t,i} = -\rho - 2\underline{\kappa}W_{t,ii} - \bar{\kappa}\sum_{j=1, j\neq i}^{n}(|W_{t,ij}| + |W_{t,ji}|) - \tau$ for any $\tau > 0$.*

## 3.2 EFFICIENT IMPLEMENTATION OF REGULARIZERS FOR CONVOLUTIONAL LAYERS

NNs are widely employed to perform image classification tasks, and convolutional layers have proved to be effective for image processing. However, convolution operations on inputs $x_t$ are usually not given in the form of $W_t x_t + b_t$ appearing in (8), which hampers the direct use of the regularizers described in Section 3.1. Although the convolution operation can be represented as $W_t x_t + b_t$ due to its linearity property, it might be burdensome to obtain $W_t$. Hence, we propose a new regularizer directly defined on the convolution filters to avoid computing $W_t$.

By construction, the input $x_0$ and the output $x_T$ of the NODE (1) have the same size. Therefore, we consider convolution operations (Goodfellow et al., 2016) that preserve the dimension of the input. Suppose the size of the input $X$ and the output $Y$ of the convolution operation is $D \times P \times H$, where $P$ and $H$ are the width and height, respectively, of the image, and $D$ is the number of channels. Let $X_i$ be the $i$-th channel of the input $X$, and $Y_j$ be the $j$-th channel of the output $Y$. Both channels have size $P \times H$. Furthermore, let the filters of the convolution operations be $C_i^j$, $i, j = 1, \ldots, D$, where $C_i^j$ represents the filter map from the $i$-th input channel to the $j$-th output channel. Since inputs and outputs of NODEs have the same size, the convolution operations must satisfy additional conditions. For example, if the filter $C_i^j$ is of size $3 \times 3$, the input size can be preserved by adding a zero-padding of 1 to the input and by applying a stride of 1 (Ciccone et al., 2018). The convolution operation can be written as

$$Y_j = \sum_{i=1}^{D} C_i^j * X_i, \quad \forall j \in \{1, \ldots, D\}, \tag{13}$$

where $*$ denotes the convolution operator. Let $\mathrm{Vec}(X)$ be the column vector concatenating the transpose of all the rows of $X_i$ for all $i$. Then, (13) can be written as

$$\mathrm{Vec}(Y) = W \times \mathrm{Vec}(X),$$

for some weight matrix $W \in \mathbb{R}^{n \times n}$, where $n = D \times P \times H$. From (13), we can see that every element of $W$ is a linear function of $C_i^j$. However, computing $W$ from $C_i^j$ can be time-consuming. The following lemma reveals important connections between the matrix $W$ and the filters $C_i^j$, that can be leveraged to directly regularize the filters $C_i^j$ for imposing contractivity.

**Lemma 1** (Ciccone et al. (2018)). *Suppose the size of $C_i^j$ is $3 \times 3$, and the convolution operation is applied with a zero-padding of $1$ and a stride of $1$. Then the following results hold.*

*(1) Let $\{C_d^d\}_{\mathrm{center}}$ denote the center element of $C_d^d$, $d = 1, \ldots, D$. Then*

$$W_{ii} = \{C_d^d\}_{\mathrm{center}}, \quad i = P \times H \times (d-1) + 1, \ldots, P \times H \times d,$$

*(2) Let $\{C_j^d\}_{kl}$ denote the $kl$-th elements of $C_j^d$. Then,*

$$\sum_{j=1}^{n} |W_{ij}| \leq \sum_{j=1}^{D} \sum_{k,l} |\{C_j^d\}_{kl}|, \quad i = P \times H \times (d-1) + 1, \ldots, P \times H \times d,$$

$$\sum_{j=1}^{n} |W_{ji}| \leq \sum_{j=1}^{D} \sum_{k,l} |\{C_d^j\}_{kl}|, \quad i = P \times H \times (d-1) + 1, \ldots, P \times H \times d.$$

**Remark 6.** *Although Lemma 1 only considers convolution operations with a filter size $3 \times 3$, a zero-padding of $1$ and a stride of $1$, the result can also be extended to other convolution operations that preserve the size of the input, for example, the convolution operation with a filter size $5 \times 5$, a zero-padding of $2$ and a stride of $1$. For more details, please refer to Ciccone et al. (2018).*

In view of Lemma 1, for $i = P \times H \times (d-1) + 1, \ldots, P \times H \times d$, we have

$$\sum_{j=1, j \neq i}^{n} (|W_{ij}| + |W_{ji}|) = \sum_{j=1}^{n} (|W_{ij}| + |W_{ji}|) - 2|W_{ii}|$$

$$\leq \sum_{j=1}^{D} \left( \sum_{k,l} |\{C_j^d\}_{kl}| + \sum_{k,l} |\{C_d^j\}_{kl}| \right) - 2|\{C_d^d\}_{\mathrm{center}}|. \tag{14}$$

Therefore, if the NN in (8) contains a convolutional layer with filters $C_i^j$, one can use the following regularization term

$$
\text{reg}\left(\{C_i^j\}_{i,j=1}^D\right) =
$$
$$
\sum_{d=1}^D P \times H \times \text{ReLU}\left(\rho + 2(\underline{\kappa} + \bar{\kappa})\{C_d^d\}_{\text{center}} + \bar{\kappa}\sum_{j=1}^D\left(\sum_{k,l}|\{C_j^d\}_{kl}| + \sum_{k,l}|\{C_d^j\}_{kl}|\right)\right),
$$
(15)

which is based on the expression of $W_{ii}$ in Lemma 1, the upper bound (14), the contractivity requirement (9), and the constraint $W_{ii} < 0$.

**Remark 7.** *The regularizer (15) includes the coefficient $P \times H$, which usually is very large for image classification tasks. In experiments of Section 4, we omit the term $P \times H$ in (15), and embed it into the regularization parameter $\gamma$.*

## 4 EXPERIMENTS

In this section, first, we compare the pros and cons of the proposed regularizers, and second, we empirically validate the improvement in the robustness of convolutional NODEs against different forms of input noise and adversarial attacks by using the contractivity-promoting regularizers on MNIST and FashionMNIST classification tasks.

### 4.1 COMPARISONS OF DIFFERENT REGULARIZATION TERMS

In Section 3, we proposed two different approaches for promoting contraction through regularization. The first one exploits the regularization term (7), whereas, the second one focuses on the NODEs (8), and utilizes the regularization term (12). Both methods have pros and cons. (12) is more computationally efficient, but might affect the representation power of the NODEs. Indeed (12) aims to make (9) and further (6) hold for all $x \in \mathbb{R}^n$. In contrast, the regularizer (7) only promotes contractivity constraints on the sampled states $x_k^i$. As a result, the trained NODE is expected to be contractive only in the neighborhood of the points $x_k^i$. This property may be beneficial for some learning problems as illustrated in the following example.

Consider the learning task shown in Figure 1a, where the goal is to train a NODE to learn a map associating $x_0^1$ to $x_T^1$ and $x_0^2$ to $x_T^2$. Since the distance between $x_0^1$ and $x_0^2$ is smaller than the distance between $x_T^1$ and $x_T^2$, we cannot obtain a satisfactory globally contractive NODE, that is, a NODE satisfying the contractivity condition (6). Instead, good performance can be achieved by a NODE, which fulfills the contractivity condition (7) involving only the sampled states. The learned trajectory and the flow of the NODE trained with the regularization term (7) are shown in Fig. 1b. To demonstrate the contractivity properties of the trained NODE, we sample the blue circles around $x_0^1$ and $x_0^2$ (see Fig. 1b), and plot the sets corresponding to the NODE outputs in red. We can observe that the area of the red regions is smaller than the area of the input circles, which is expected from local contractivity.

### 4.2 MNIST AND FASHIONMNIST CLASSIFICATION TASKS

We evaluate the performance of the proposed regularization schemes on image classification tasks for the MNIST and FashionMNIST datasets, which are based on images of size $28 \times 28$. In both cases, we use the NODE (8) with convolutional layers. We train both a vanilla NODE (i.e., using $\gamma = 0$ in (3)) and the NODE with the regularization term (15), which we refer to as contractive NODE (CNODE), for ten different seeds so obtaining 10 versions of each model.

The NODE structure is described as follows, where unless otherwise specified, the same parameters are used for both the MNIST and the FashionMNIST datasets. First, the image is processed by $h_\alpha(\cdot)$, which is a convolution operation with a filter size $3 \times 3$, a stride of 1, and a channel number of 8 and 16 for MNIST and FashionMNIST dataset, respectively. Second, it is processed by the NODE (8) for $T = 0.1$, where the NN is also a convolution operation with a filter size $3 \times 3$, a zero-padding of 1, and a stride of 1. We use FE discretization with step size $h = 0.01$ for training the NODEs.

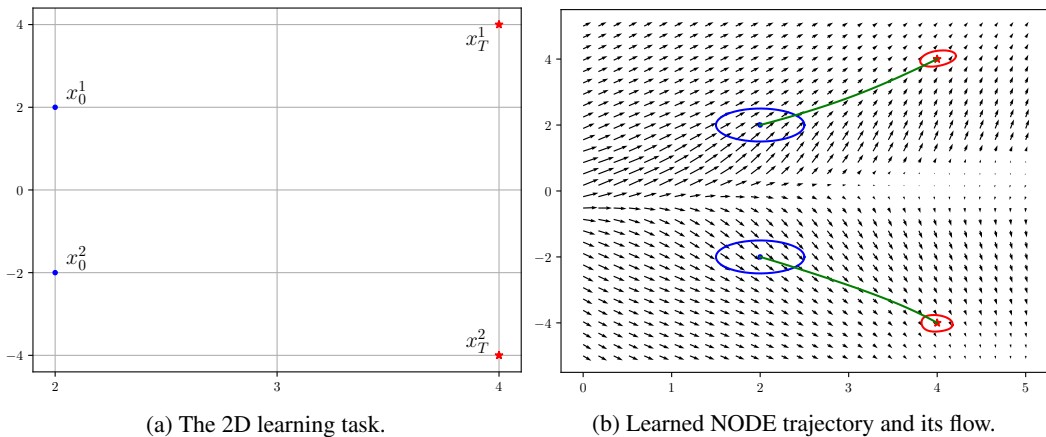

(a) The 2D learning task.   (b) Learned NODE trajectory and its flow.

Figure 1: Example of a simple NODE trained with the contractivity-promoting regularizer (7).

Finally, the output of the NODE is followed by a fully connected layer $g_\beta(\cdot)$ with output dimension 10. Due to the smoothness requirement of $f$ in (1) and the slope restrictions, we select the activation function in (8) to be the smooth leaky ReLU function, given by $\sigma(x) = 0.1x + 0.9\log(1 + e^x)$, which satisfies $0.1 \leq \sigma'(\cdot) \leq 1$. We use the Adam optimizer to minimize the cross-entropy loss. The initial learning rate for the Adam optimizer is $0.05$, and the learning rate is reduced by a factor of $0.7$ after every training epoch. The maximal number of training epochs is 20. For the regularizer (15), we use $\rho = 2$. The weight $\gamma$ for the regularization term (15) is set to 1. The contraction rate $\rho$ and the regularization parameter $\gamma$ are selected using grid search. We show in Appendix A.2 and Appendix A.3 that the average test accuracy is quite insensitive to the choice of $\rho$ and $\gamma$. Moreover, we change the convolution parameters in the NODE and repeat the experiment. The results are shown in Appendix A.4, which implies that with different convolution parameters, we can still achieve improved robustness performance with contractivity regularization.

We test the performance of the vanilla NODE and CNODE against noisy test datasets, where the images are perturbed by zero mean Gaussian noise, and salt&pepper noise (Schott et al., 2019). For each kind of noise, we generate several noisy test datasets with different noise strengths. Moreover, we test the adversarial robustness of the NODEs with respect to fast-gradient-sign-method (FGSM) (Goodfellow et al., 2014) and projected gradient descent (PGD) attacks (Madry et al., 2017). Tables 3 and 4 summarize the mean and standard deviations of the classification accuracy over all test sets. In Table 3, $\sigma$ is the standard deviation of the Gaussian noise and $\epsilon$ denotes the proportion of image pixels corrupted by the impulse noise. The results of robustness against adversarial attacks are reported in Table 4, where $\delta$ represents the $l_\infty$ amplitude of perturbations in FGSM and PGD attacks. The best performance in each column appears in **bold**. To give an idea of the intensity of perturbations, we provide samples of test images in Appendix A.1.

|  | No Noise | Gaussian | | | Salt&Pepper | | |
|---|---|---|---|---|---|---|---|
| **MNIST** | $\sigma = \epsilon = 0$ | $\sigma = 0.1$ | $\sigma = 0.2$ | $\sigma = 0.3$ | $\epsilon = 0.1$ | $\epsilon = 0.2$ | $\epsilon = 0.3$ |
| Vanilla NODE | 98±0.3 | 65±23 | 45±21 | 37±16 | 76±9 | 54±11 | 42±8 |
| CNODE | 98±0.1 | **94±4** | **79±8** | **62±12** | **88±4** | **68±8** | **48±8** |

| **FashionMNIST** | $\sigma = \epsilon = 0$ | $\sigma = 0.1$ | $\sigma = 0.2$ | $\sigma = 0.3$ | $\epsilon = 0.1$ | $\epsilon = 0.2$ | $\epsilon = 0.3$ |
|---|---|---|---|---|---|---|---|
| Vanilla NODE | 88±0.1 | 75±4 | 47±4 | 35±4 | 69±2 | 51±4 | 38±5 |
| CNODE | 88±0.2 | **85±1** | **72±2** | **55±4** | **75±2** | **57±5** | **42±5** |

Table 1: Classification accuracy over noisy test images (mean ± standard deviation).

From the tables, we can observe that the CNODEs achieve higher mean classification accuracy than the vanilla NODEs in the presence of image perturbations. In some cases, the performance improvements are very significant (up to 34% for the case of Gaussian noises). Moreover, the standard deviations with CNODEs are either the same or less than those with vanilla NODEs in

|  | FGSM | | | PGD | | |
|---|---|---|---|---|---|---|
| **MNIST** | $\delta = 0.01$ | $\delta = 0.02$ | $\delta = 0.03$ | $\delta = 0.01$ | $\delta = 0.02$ | $\delta = 0.03$ |
| Vanilla NODE | 92±2 | 67±8 | 42±9 | 91±3 | 63±12 | 36±11 |
| CNODE | **95±0.4** | **86±2** | **68±4** | **95±0.4** | **86±2** | **66±4** |
| **FashionMNIST** | $\delta = 0.01$ | $\delta = 0.02$ | $\delta = 0.03$ | $\delta = 0.01$ | $\delta = 0.02$ | $\delta = 0.03$ |
| Vanilla NODE | 63±1 | 31±1 | 13±1 | 62±1 | 29±1 | 11±1 |
| CNODE | **72±1** | **49±2** | **28±2** | **71±1** | **47±3** | **26±2** |

Table 2: Classification accuracy over adversarial attacks (mean ± standard deviation).

almost all the experiments, which means, CNODEs are less sensitive than vanilla NODEs to the selection of initialization seeds.

## 5  CONCLUSIONS

In this paper, we use contraction from dynamical system theory to improve the robustness of NODEs. We propose regularizers with different degrees of flexibility and different computational requirements to promote contractivity. The good performance of the resulting NNs is illustrated on image classification tasks. Future work will focus on the development of easy-to-compute regularizers for classes of NODEs stemming from specific choices of $f$ in (1).

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

# A  APPENDIX

## A.1  CLASSIFICATION EXPERIMENTS: EXAMPLES OF PERTURBED IMAGES

In Figure 2, we provide samples of the perturbed MNIST and FashionMNIST datasets that have been used for testing the performance of different NODEs in Section 4. For the meaning of the parameters $\sigma, \epsilon$ and $\delta$, capturing the perturbation magnitude, we defer the reader to Section 4.2.

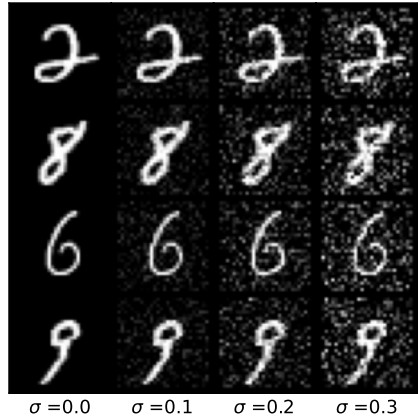

$\sigma = 0.0 \quad \sigma = 0.1 \quad \sigma = 0.2 \quad \sigma = 0.3$

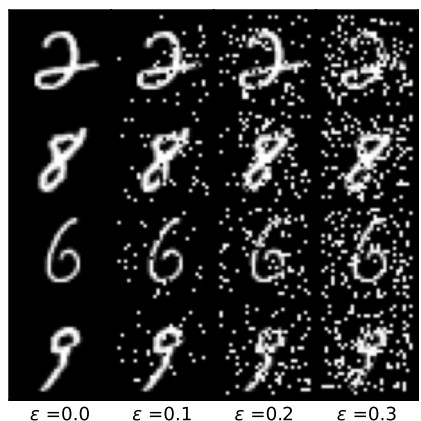

$\varepsilon = 0.0 \quad \varepsilon = 0.1 \quad \varepsilon = 0.2 \quad \varepsilon = 0.3$

(a) MNIST samples perturbed by Gaussian noise.     (b) MNIST samples perturbed by salt&pepper noise.

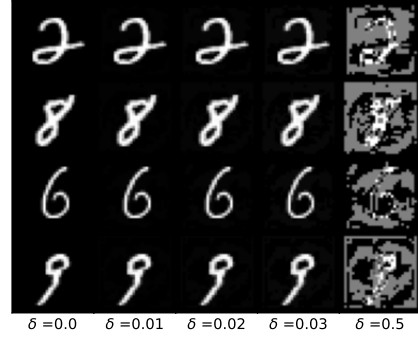

$\delta = 0.0 \quad \delta = 0.01 \quad \delta = 0.02 \quad \delta = 0.03 \quad \delta = 0.5$

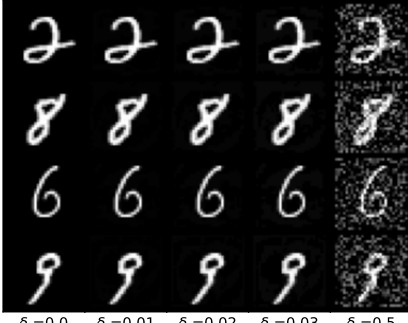

$\delta = 0.0 \quad \delta = 0.01 \quad \delta = 0.02 \quad \delta = 0.03 \quad \delta = 0.5$

(c) MNIST samples perturbed by FGSM attacks.     (d) MNIST samples perturbed by PGD attacks.

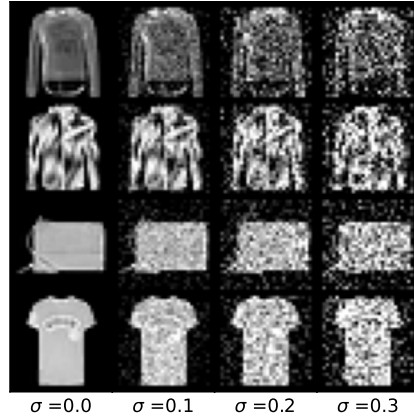

$\sigma = 0.0 \quad \sigma = 0.1 \quad \sigma = 0.2 \quad \sigma = 0.3$

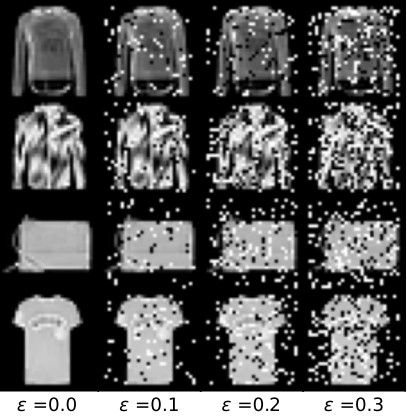

$\varepsilon = 0.0 \quad \varepsilon = 0.1 \quad \varepsilon = 0.2 \quad \varepsilon = 0.3$

(e) FashionMNIST samples perturbed by Gaussian noise.     (f) FashionMNIST samples perturbed by salt&pepper noise.

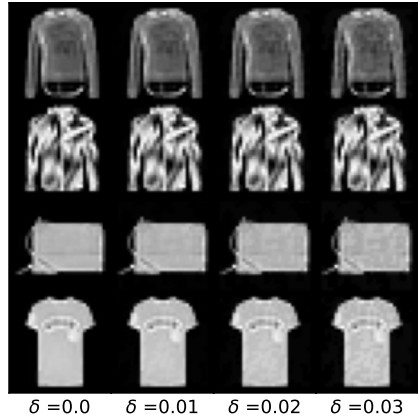 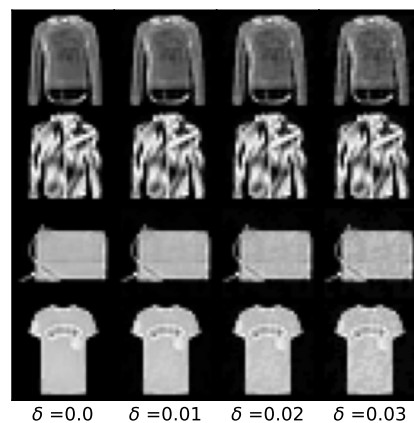

$\delta = 0.0$ $\quad$ $\delta = 0.01$ $\quad$ $\delta = 0.02$ $\quad$ $\delta = 0.03$ $\qquad\qquad$ $\delta = 0.0$ $\quad$ $\delta = 0.01$ $\quad$ $\delta = 0.02$ $\quad$ $\delta = 0.03$

(g) FashionMNIST samples perturbed by FGSM attacks. (h) FashionMNIST samples perturbed by PGD attacks.

Figure 2: Examples of perturbed images in MNIST and FashionMNIST classification tasks

## A.2 CONTRACTION RATE VS CLASSIFICATION ACCURACY

In this appendix, we analyze how the contraction rate affects the classification accuracy. For this purpose, we use the MNIST dataset and images perturbed by Gaussian noises or FGSM attacks. We use contraction rates $\rho$ in the set $\{0.1, 2, 5, 7, 10, 12, 15\}$, train the CNODEs, and obtain 10 models for each $\rho$ by using different seeds. Then, we calculate the classification accuracy of these models on the clean test dataset, the test dataset perturbed by Gaussian noises, and the test dataset attacked by FGSM. The mean and the standard deviations of the classification accuracy are plotted in Figure 3, Figure 4 and Figure 5, where the solid line represents the mean and the shaded region spans one standard deviation on each side of the mean. We can observe that the average classification accuracy does not vary significantly for different contraction rates. This suggests that the choice of the contraction rate is not critical for the MNIST experiments discussed in Section 4.

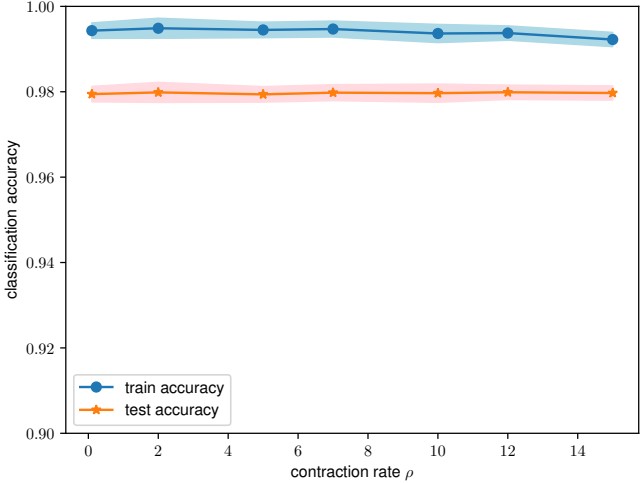

Figure 3: Classification accuracy on the clean test dataset with respect to different contraction rates $\rho$.

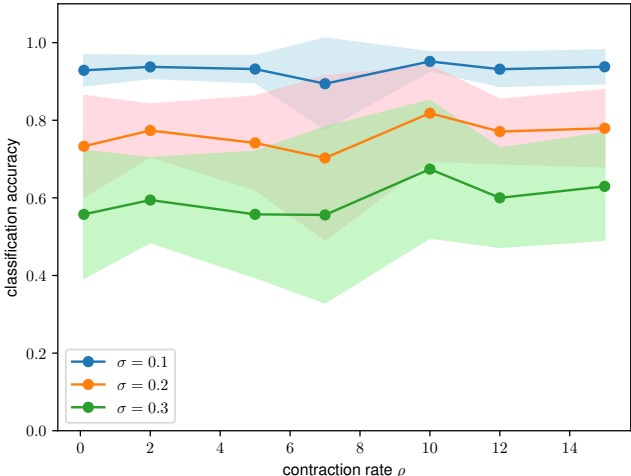

Figure 4: Classification accuracy on test dataset perturbed by Gaussian noise with respect to different contraction rates $\rho$.

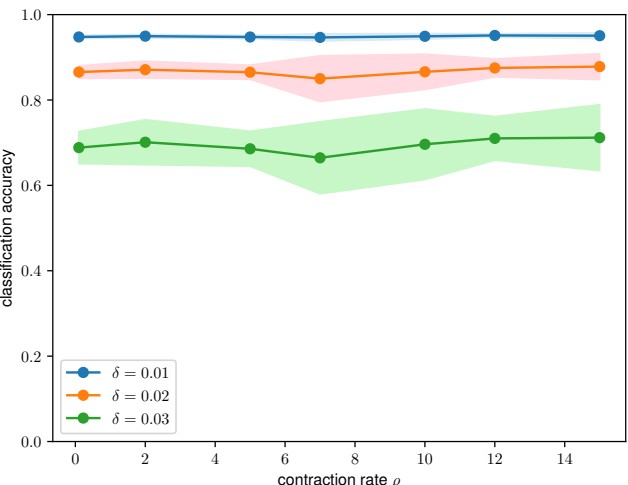

Figure 5: Classification accuracy on test dataset perturbed by FGSM attacks with respect to different contraction rates $\rho$.

## A.3 REGULARIZER WEIGHT VS CLASSIFICATION ACCURACY

In this appendix, we analyze how the regularizer weight affects the classification accuracy. For this purpose, we use the MNIST dataset and images perturbed by Gaussian noises or FGSM attacks. We use regularization parameter $\gamma$ in the set $\{0.1, 1, 5, 10, 20, 30, 40, 50\}$, train the CNODEs, and obtain 10 models for each $\gamma$ by using different seeds. Then, we calculate the classification accuracy of these models on the clean test dataset, the test dataset perturbed by Gaussian noises, and the test dataset attacked by FGSM. The mean and the standard deviations of the classification accuracy are plotted in Figure 6, Figure 7, and Figure 8, where the solid line represents the mean and the shaded region spans one standard deviation on each side of the mean. We can observe that the average classification accuracy does not vary significantly with different values of $\gamma$. Therefore, the average

robustness performance does not rely heavily on the selection of the regularization parameter $\gamma$ in the MNIST experiment in Section 4.

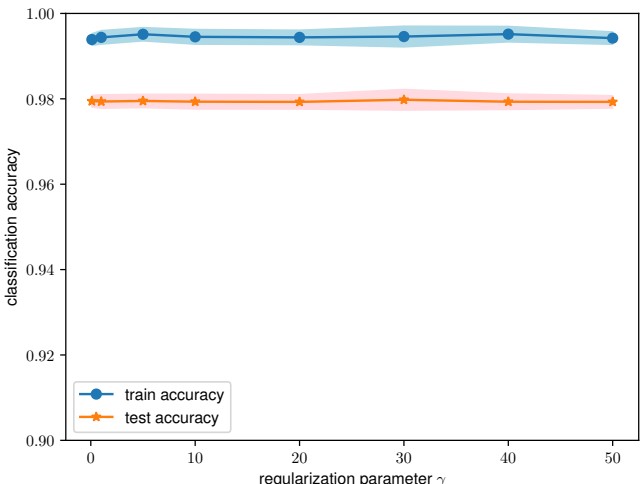

Figure 6: Classification accuracy on the clean test dataset with respect to different regularization parameters $\gamma$.

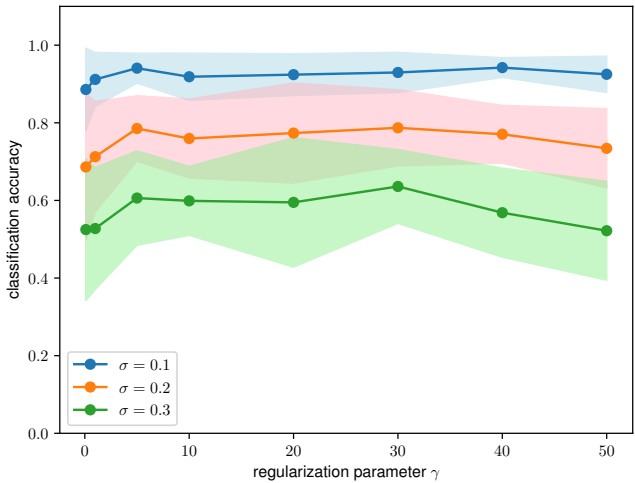

Figure 7: Classification accuracy on test dataset perturbed by Gaussian noise with respect to different regularization parameters $\gamma$.

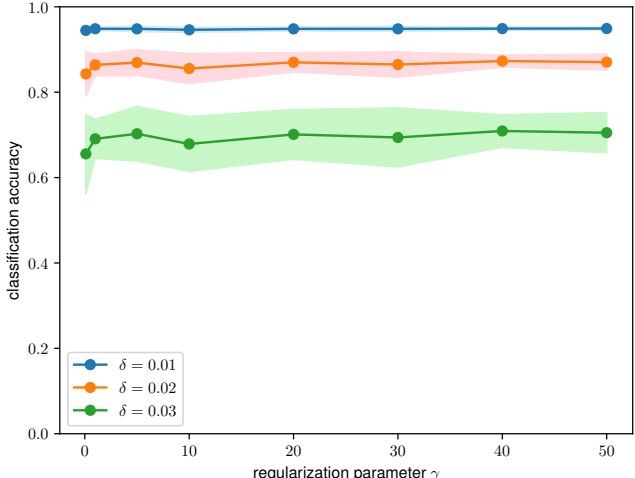

Figure 8: Classification accuracy on test dataset perturbed FGSM attacks with respect to different regularization parameters $\gamma$.

## A.4  CONVOLUTION PARAMETERS VS CLASSIFICATION ACCURACY

In this Appendix, we perform an ablation study by selecting different convolution parameters and demonstrate that CNODEs can still achieve improved robustness performance. We set parameters of the convolution operation in the NODE to the following two groups:

- Group 1: a filter size $5 \times 5$, a zero-padding of 2, and a stride of 1.
- Group 2: a filter size $7 \times 7$, a zero-padding of 3, and a stride of 1.

Then we re-conduct the experiment. The maximal number of training epochs is 30. The other training parameters are set to be the same with those in Section 4.2. The average test accuracy and the standard deviation data are shown in the following table, where CNODE(5) and CNODE(7) represent the CNODE with convolution parameter Group 1 and Group 2, respectively. We can observe that with different convolution parameters, the CNODE can still achieve improved performance.

|  | No Noise | Gaussian | | | Salt&Pepper | | |
|---|---|---|---|---|---|---|---|
| **MNIST** | $\sigma = \epsilon = 0$ | $\sigma = 0.1$ | $\sigma = 0.2$ | $\sigma = 0.3$ | $\epsilon = 0.1$ | $\epsilon = 0.2$ | $\epsilon = 0.3$ |
| Vanilla NODE | 98±0.3 | 65±23 | 45±21 | 37±16 | 76±9 | 54±11 | 42±8 |
| CNODE(5) | 98±0.2 | **95±2** | **81±7** | **65±8** | **87±3** | **66±4** | **45±4** |
| CNODE(7) | 98±0.1 | **95±2** | **81±7** | **65±11** | **88±3** | **67±6** | **46±6** |
| | | | | | | | |
| **FashionMNIST** | $\sigma = \epsilon = 0$ | $\sigma = 0.1$ | $\sigma = 0.2$ | $\sigma = 0.3$ | $\epsilon = 0.1$ | $\epsilon = 0.2$ | $\epsilon = 0.3$ |
| Vanilla NODE | 88±0.1 | 75±4 | 47±4 | 35±4 | 69±2 | 51±4 | 38±5 |
| CNODE(5) | 88±0.1 | **83±1** | **65±4** | **48±6** | **75±3** | **58±6** | **44±7** |
| CNODE(7) | 88±0.1 | **83±1** | **61±5** | **42±5** | **75±2** | **55±5** | **40±5** |

Table 3: Classification accuracy over noisy test images (mean $\pm$ standard deviation).

| | FGSM | | | PGD | | |
|---|---|---|---|---|---|---|
| **MNIST** | $\delta = 0.01$ | $\delta = 0.02$ | $\delta = 0.03$ | $\delta = 0.01$ | $\delta = 0.02$ | $\delta = 0.03$ |
| Vanilla NODE | 92±2 | 67±8 | 42±9 | 91±3 | 63±12 | 36±11 |
| CNODE(5) | **95±0.5** | **87±3** | **69±6** | **95±0.6** | **86±3** | **67±6** |
| CNODE(7) | **95±0.5** | **87±2** | **70±6** | **95±0.5** | **87±3** | **69±6** |
| **FashionMNIST** | $\delta = 0.01$ | $\delta = 0.02$ | $\delta = 0.03$ | $\delta = 0.01$ | $\delta = 0.02$ | $\delta = 0.03$ |
| Vanilla NODE | 63±1 | 31±1 | 13±1 | 62±1 | 29±1 | 11±1 |
| CNODE(5) | **71±1** | **46±3** | **25±3** | **71±1** | **44±3** | **23±3** |
| CNODE(7) | **73±1** | **48±2** | **27±2** | **72±1** | **47±2** | **25±2** |

Table 4: Classification accuracy over adversarial attacks (mean ± standard deviation).

