# OpenReview forum: "Robust Neural ODEs via Contractivity-promoting Regularization"
_ICLR.cc/2023/Conference — Submitted to ICLR 2023_

### Official Review · Reviewer_pw1W · 2022-10-24

**Confidence:** 4
**Correctness:** 2
**Technical Novelty And Significance:** 2
**Empirical Novelty And Significance:** 2
**Recommendation:** 3

**Clarity, Quality, Novelty And Reproducibility:**

Overall, the paper is clearly written, and the proofs are correct. The proposed efficient regularizers for a special neural ODE is novel. However, it does not provide robust classifiers (see point 1) in Weakness).
There are also some minor typos in the theorems. For instance, in Lemma 1, the subscript t is missing.

**Strength And Weaknesses:**

Strength:
1) Leveraging control tools to enforce contraction properties on neural ODEs is an important direction that brings new insights into the learning community.
2) For neural ODEs whose dynamics is one activation function composed with one linear layer, the authors proposed an efficient way to enforce the contraction property.

Weaknesses:
1) My main concern for this paper is: Enforcing global contractivity can NOT provide robust classifiers. Indeed, it can provide robustness, but robust classification requires classifying images into the correct classes, not into one class. Following Def 1, if global contraction holds for a neural ODE, then with long enough integration time, all x_t will converge to a single point no matter what x_0 it starts with. That means, a global contractive neural ODE classifier will classify all images into one class, which loses classification capability.
A neural ODE that satisfies (15) should classify images into one class. However, the experiments still show nontrivial accuracy. I suspect that it is because the integration time is not long enough, or the contraction rate is very small.
2) Some recent papers (e.g. [1]) shows that adversarial robustness improvements of neural ODEs may be from obfuscated gradients. Therefore, the evaluation of the robustness improvement should be evaluated with AutoAttack (https://github.com/fra31/auto-attack), and the accuracy after each attack should be reported so that it provides more info on whether the robustness improvement is due to gradient obfuscation. Or, the authors can exploit whether they can provide a certification for the robustness, if the contraction property holds everywhere, they should be able to evaluate the Lipschitz of the neural ODE and certify the robustness.
3) The efficient regularizer sacrifices the expressiveness of neural ODEs in two folds: one layer dynamics, and the lower and bounds when deriving the regularizer. The authors could potentially explore whether they can relax the global contraction requirement (also see point 1)) so that they can enable the dynamics to be more expressive.

[1] Yifei Huang et al. “Adversarial Robustness of Stabilized Neural ODE Might be from Obfuscated Gradients”. In: Mathematical and Scientific Machine Learning. PMLR. 2022, pp. 497–515.

**Summary Of The Paper:**

This paper proposes regularization terms to promote the contractivity of neural ODEs. It derives a computational efficient regularizer for a special class of neural ODEs and showed empirical robustness comparing with plain neural ODEs.

**Summary Of The Review:**

This papers proposes regularization terms to enforce contraction properties in neural ODEs with a goal to improve the robustness of neural ODEs. However, in many applications such as image classification, robustness does not mean to have all inputs to converge to a single point. Therefore, global contraction may not be a desired property to have. I encourage the authors to explore "local" contraction for each class of images, or other applications where global contraction is desired.

---

> ### Author Response · Authors · 2022-11-18
> **Response to the Reviewer pw1W**
>
> We thank the Reviewer for reading the paper carefully and providing a valuable feedback. We refer the Reviewer to the following point-wise reply.
>
> ## Reply to the Weakness 1
>
> Firstly, we provide two alternating methods for promoting contractivity with different computational requirements.  The regularizer (7) promotes the contractivity locally, whereas the regularizers (12), and (15) promote the global contractivity of a NODE. Secondly, while training NODEs with globally contractive regularizers, the depth parameter $T$ is finite, and we can also tune the contraction rate $\rho$ to set the Lipschitz constant $e^{-\rho T}$ as desired. As a result, the NODE trajectory would not converge to the same point during training, which ensures good learning performance. The reviewer can also refer to Figure 1 of (Zakwan et al. 2023) for an illustrative example showing that global contraction does not prevent from obtaining good learning results.
>
> We have added Remark 3 in the revised manuscript to address the reviewer’s concerns.
>
> - [Zakwan et al. 2023] M. Zakwan, L. Xu, and G. Ferrari-Trecate, “Robust Classification Using Contractive Hamiltonian Neural ODEs,” *IEEE Control Systems Letters*, vol. 7, pp. 145–150, 2023.
>
> ## Reply to the Weakness 2
>
> We have provided a certificate for robustness in terms of the Lipschitz bound of the input-output map of the NODE. Specifically, we have shown in Section 2.2 that the Lipschitz constant of the input-out map is $e^{-\rho T}$, which is smaller than 1. Therefore, the neural ODE is certified to be robust to perturbations.
> We tried to perform Autoattacks using the standard library of torchattacks. However, it was taking several days with our current computational resources, and we could not achieve any results in this period. Moreover, from the lens of contraction theory, all possible perturbations in a ball centered around the training points experience a diminishing effect. Hence, for different methods of attacks, we expect similar results as for the FGSM and PDG attacks.
> Moreover, as mentioned in [1], the obfuscated gradients of NODEs come from using adaptive step-size solvers (e.g., Bosh3, adaptive Heun, and Dopri5) during the forward inference. However, in this work, we used a fixed-step size method (i.e. Euler) for training the NODEs. Therefore, we expect that for our architectures, the robustness is coming from the contraction, not the obfuscated gradients.
>
> - [1] Yifei Huang et al. “Adversarial Robustness of Stabilized Neural ODE Might be from Obfuscated Gradients”. *Mathematical and Scientific Machine Learning*. PMLR, pp. 497–515,  2022.
>
> ## Reply to the Weakness 3
>
> We have provided regularizers (see (7)) to promote local contractivity. However, they might be computationally expensive to optimize. Moreover, we are using time-varying NODEs, which are more expressive than vanilla time-invariant NODE. Indeed, after discretization, our models are equivalent to residual networks with different parameters in each layer. Finally, as explained in the reply to point 1, while global contractivity might reduce the expressivity, it does not necessarily harm the nominal test accuracy.

---

### Official Review · Reviewer_MdaE · 2022-10-25

**Confidence:** 3
**Correctness:** 3
**Technical Novelty And Significance:** 4
**Empirical Novelty And Significance:** 4
**Recommendation:** 5

**Clarity, Quality, Novelty And Reproducibility:**

This paper is well written and the organization is satisfied. The method is theoretically induced and computationally efficient. The empirical results seem to validate that the empirical contribution of the proposed method is significant.

**Strength And Weaknesses:**

Strength
+ The proposed method is theoretically induced. The authors make the contractivity-promoting regularization computationally cheap based on theoretical results, thus enabling the authors to apply contractivity-promoting regularization to neural ODE in practice.
+ The empirical results support the claim. Compared to vanilla neural ODE, the proposed contractive neural ODE significantly improve the robust accuracy.

Weaknesses
+ The adversarial robustness evaluation could be unreliable. This paper [1] has pointed out a reliable method to evaluate the robustness of neural ODE. It would be better for the authors to evaluate the performance and compare the proposed method with previous methods such as [2,3].
+ It would be better to provide an ablation study on the kernel size of the CNN filter, to validate the proposed method can be effective on different neural structures.


[1] Evaluating the Adversarial Robustness of Adaptive Test-time Defenses. Croce et al. ICML 2022.
[2] Stable Neural ODE with Lyapunov-Stable Equilibrium Points for Defending against Adversarial Attacks. Kang et al. 2021.
[3] On robustness of neural ordinary differential equations. Yan et al. ICLR 2019.


**Summary Of The Paper:**

This paper leverages contraction theory to improve the robustness of neural ordinary differential equation (ODE). Directly obtaining contraction property needs to regularize the Jacobian matrix, which is computationally heavy. The authors theoretically prove that penalizing the weight matrix can be a contractivity-promoting regularization under a mild assumption that the slope of the activation function of neural ODE is bounded. The experimental results validate the effectiveness of contractive neural ODEs in enhancing robustness against corruption and adversarial attacks.

**Summary Of The Review:**

This paper uses the contraction theory to improve the robustness of neural ODEs. The application of the contraction theory to neural ODE is non-trivial. Thus, I believe the proposed method is novel. However, I have some concerns regarding the empirical results. I would like to rise my score if the authors solve my concerns.

---

> ### Author Response · Authors · 2022-11-18
> **Response to the Reviewer MadE**
>
> We thank the Reviewer for reading the paper carefully and providing a valuable feedback. We refer the Reviewer to the following point-wise reply.
>
> ## Reply to the Weakness 1
>
> We used the standard library of “torchattacks” to implement Autoattacks for our architecture. However, a single experiment was taking several days with our available computational resources. Because of the strict time constraints of the review process, we could not implement the robustness evaluation methods in [1]. Moreover, from the lens of contraction theory, all possible perturbations in a ball centered around the training points experience a diminishing effect. Hence, for different methods of attacks, we expect similar results as for the FGSM and PDG attacks.  Besides, we use different neural network architectures as compared with [2] and [3], that also have complex feature extractors and output layers. The good robustness performance in these works might come from these complicated architectures. In comparison, not to blur the main picture, we only use a single layered feature extractor and an output layer.  Moreover, the authors of [2] did not release the full code for their simulations, making the comparison with their results difficult.
>
> - [1] Evaluating the Adversarial Robustness of Adaptive Test-time Defenses. Croce et al. ICML 2022.
> - [2] Stable Neural ODE with Lyapunov-Stable Equilibrium Points for Defending against Adversarial Attacks. Kang et al. 2021.
> - [3] On robustness of neural ordinary differential equations. Yan et al. ICLR 2019.
>
> ## Reply to the Weakness 2
>
> We have changed the kernel size of the CNN filter and re-conduct the simulations, please see Appendix 4 of the revised manuscript. The result shows that with different kernel sizes, CNODEs still have good robustness performances.

---

### Official Review · Reviewer_8Xx6 · 2022-11-04

**Confidence:** 5
**Correctness:** 2
**Technical Novelty And Significance:** 2
**Empirical Novelty And Significance:** 2
**Recommendation:** 3

**Clarity, Quality, Novelty And Reproducibility:**

The paper is generally clear, however it does not reach the quality for publication (see weaknesses discussed below).

**Strength And Weaknesses:**

Strengths:

1. The robustness of neural ODE is an important topic and this paper brings some new insights into this problem.

2. The contraction theory inspired weight regularization based approach for enhancing the robustness of ODE is new, although it is not well evaluated and its connection to previous Lyapunov theory based approaches are not thoroughly discussed.


Weaknesses:

1. The way of training neural ODEs in this paper is naive via unrolling the steps via time discretization. This is not the best way to train neural ODEs and has great impact on training cost and practicability.

2. Many theoretical results presented in this paper are not new (e.g., Lemma 1, Lemma 2). They should not take too much space. The main contribution seems to be an application of Gersgorin disk theorem for the weight matrices of neural ODEs, and it is arguably a significant contribution.

3. Empirical evaluations are very weak, with MNIST and Fashion-MNIST only. These are not representative datasets, since MNIST-like datasets are too simple and do not reflect true performance of an algorithm under any more practical datasets. In addition, a few important baselines which also focus on training robust ODE are missing, see [1][2]. [1][2] use an elegant and fundamental approach from the Lyapunov theory.

Question:

If a NODE is fully contractive then it is possible that all inputs are contracted to the same output. Thus for any real tasks it can only be locally contractive (e.g., a point very close to x_0 should lead to the same output). The main theory is independent of x_0, so it is for global contraction. How can you prevent that the output of NODE for all input points collapse to the same value?

[1] Rodriguez, Ivan Dario Jimenez, Aaron Ames, and Yisong Yue. "LyaNet: A Lyapunov framework for training neural ODEs." International Conference on Machine Learning. PMLR, 2022.
[2] Kang, Qiyu, et al. "Stable neural ode with lyapunov-stable equilibrium points for defending against adversarial attacks." Advances in Neural Information Processing Systems 34 (2021): 14925-14937.

**Summary Of The Paper:**

This paper studies the robustness of neural ODEs, by studying the condition when neural ODEs are contractive, i.e., the trajectories of the ODE converge to the same value exponentially fast. Based on existing lemmas on the condition of contraction, the authors propose a weight regularization based approach during training to encourage contraction of neural ODEs and thus improve its robustness. The method is demonstrated on MNIST and Fashion-MNIST datasets and its robustness under Guassian noises and adversarial attacks are better than vanilla ODEs.

**Summary Of The Review:**

This paper is not ready for publication because of the relatively weak theoretical contribution, missing related works and insufficient experiments. The robustness of neural ODEs is a very interesting topic and the proposed regularization based approach might be promising if it is thoroughly evaluated. I encourage the authors to continue working on this problem further and submit the paper to a future conference.

---

> ### Author Response · Authors · 2022-11-18
> **Response to the Reviewer 8Xx6**
>
> Thanks for reading the paper carefully and providing a valuable feedback. We refer the Reviewer to the following point-wise reply.
>
> ## Reply to Weakness 1
>
> To obtain more expressive models, our considered neural ODEs depend on time-varying parameters $\theta_t$, which is different from the model considered in (Chen et al., 2018), where time-invariant training parameters $\theta$ are used. As a result, the memory-efficient training approaches proposed in (Chen et al., 2018) might not apply to the training of our considered neural ODEs. Note also that, after discretization with the forward Euler method, the resulting models are equivalent to residual networks that can be trained even when the number of layers is high by using standard software packages. Therefore, for the sake of simplicity, we prefer the time discretization approach for training neural ODEs.
>
> - [Chen et al., 2018] T. Q. Chen, Y. Rubanova, J. Bettencourt, and D. K. Duvenaud, “Neural ordinary differential equations,”  *Advances in Neural Information Processing Systems*, pp. 6571–6583, 2018.
>
> ## Reply to Weakness 2
>
> We have revised the manuscript and stated the results in Lemma 1 (in previous version) as plain text.  However, the result in Lemma 2 (in previous version) is a variant of the result in (Ciccone et al., 2018)  and is not explicitly reported. Therefore, we prefer to keep it, also because the precise statement is needed for formulating the regularizers.
>
> - [Ciccone et al., 2018] M. Ciccone, M. Gallieri, J. Masci, C. Osendorfer, and F. Gomez, “Nais-net: Stable deep networks from non-autonomous differential equations,” *Advances in Neural Information Processing Systems*, vol. 31, 2018.
>
> ## Reply to Weakness 3
>
> We aim to show the performance of our architecture on noisy datasets without introducing ad hoc image processing layers to focus on the performance improvements brought by the new contractivity-based regularizers. Therefore, we avoided on purpose to introduce sophisticated pre- and post-layers dedicated to image processing that may influence robustness and hence blur the picture. For this reason, we do not expect our NODE to show a good performance on more complex datasets. Indeed, while using our NODE models even without regularizers (i.e., without trading off expressivity for robustness), we achieved 70 percent test accuracy on CIFAR 10 even with extensive hyperparameter tuning. However, once more, our main goal is not to best state-of-the-art methods on complex datasets but rather propose new simple and effective class of regularizers, based on contraction theory, to address robust classification.
>
> As for [1] and [2], we notice the NN architectures do use complex feature extractors and output layers and the robustness might come from these sophisticated architectures. In comparison, we only use a single convolution layer as the feature extractor and output layer.  Moreover, the authors of [2] did not release the full code for their simulations, making the comparison with their results difficult.
>
> - [1] Rodriguez, Ivan Dario Jimenez, Aaron Ames, and Yisong Yue. "LyaNet: A Lyapunov framework for training neural ODEs." *International Conference on Machine Learning*. PMLR, 2022.
> - [2] Kang, Qiyu, et al. "Stable neural ode with Lyapunov-stable equilibrium points for defending against adversarial attacks." *Advances in Neural Information Processing Systems*, 34: 14925-14937, 2021.
>
> ## Reply to the Question
>
> Firstly, we have provided approaches to ensure local contractivity, see the regularizer (7) and the discussion between the differences of local and global contractivity in Section 4.1. As pointed out in Section 4.1, the local contractivity requirement might be computationally expensive. Therefore, we have proposed regularizers based on global contractivity that are computationally more amenable.
>
> Besides, while training NODEs with globally contractivity regularizers , the depth parameter $T$ is finite, and we can also tune the contraction rate $\rho$ to set the Lipschitz constant $e^{-\rho T}$ as desired. As a result, the NODE trajectory would not converge to the same point during training, which ensures good learning performance. The reviewer can also refer to Figure 1 of (Zakwan et al. 2023) for an illustrative example showing that global contraction does not prevent from obtaining good learning results.
>
> We have added Remark 3 in the revised manuscript to address the Reviewer’s concerns.
>
> - [Zakwan et al. 2023] M. Zakwan, L. Xu, and G. Ferrari-Trecate, “Robust Classification Using Contractive Hamiltonian Neural ODEs,” *IEEE Control Systems Letters*, vol. 7, pp. 145–150, 2023.

---

### Decision · Program_Chairs · 2023-01-20

**Decision:**

Reject

**Justification For Why Not Higher Score:**

The experiments lack representative datasets, necessary baselines and comparison with previous methods.

**Justification For Why Not Lower Score:**

N/A

**Metareview: Summary, Strengths And Weaknesses:**

This paper looks for the robustness improvement of neural ordinary differential equations. For this purpose, the authors use contraction theory to design weight regularization terms, which is computationally efficient. Reviewers had a few major concerns, especially the empirical results. Reviewers argued that the experiments lack representative datasets, necessary baselines and comparison with previous methods. The rebuttal did not overcome the reviewer's objections. For these reasons, I recommend reject.